# Discovery of high-performance low-cost n-type Mg$_3$Sb$_2$-based thermoelectric materials with multi-valley conduction bands

Jiawei Zhang[1], Lirong Song[1], Steffen Hindborg Pedersen[1], Hao Yin[1,2], Le Thanh Hung[3] & Bo Brummerstedt Iversen[1]

Widespread application of thermoelectric devices for waste heat recovery requires low-cost high-performance materials. The currently available n-type thermoelectric materials are limited either by their low efficiencies or by being based on expensive, scarce or toxic elements. Here we report a low-cost n-type material, Te-doped Mg$_3$Sb$_{1.5}$Bi$_{0.5}$, that exhibits a very high figure of merit $zT$ ranging from 0.56 to 1.65 at $300-725\,$K. Using combined theoretical prediction and experimental validation, we show that the high thermoelectric performance originates from the significantly enhanced power factor because of the multi-valley band behaviour dominated by a unique near-edge conduction band with a sixfold valley degeneracy. This makes Te-doped Mg$_3$Sb$_{1.5}$Bi$_{0.5}$ a promising candidate for the low- and intermediate-temperature thermoelectric applications.

[1] Center for Materials Crystallography, Department of Chemistry and iNANO, Aarhus University, DK-8000 Aarhus, Denmark. [2] TEGnology ApS, Lundagervej 102, DK-8722 Hedensted, Denmark. [3] Department of Energy Conversion and Storage, Technical University of Denmark, 399 Frederiksborgvej, 4000 Roskilde, Denmark. Correspondence and requests for materials should be addressed to B.B.I. (email: bo@chem.au.dk).

Thermoelectric conversion technology, which can realize the direct and reversible conversion between heat and electricity without moving parts, offers a promising solution to critical energy and environmental challenges[1]. The conversion efficiency of thermoelectric devices is determined by the materials performance that is quantified by the dimensionless figure of merit, $zT = \alpha^2 \sigma T / \kappa$, where $\alpha$ is the Seebeck coefficient, $\sigma$ is the electrical conductivity, $T$ is the absolute temperature and $\kappa$ is the thermal conductivity. Numerous efforts worldwide have been directed towards the improvement of $zT$ through enhancing the power factor[2–6] ($\alpha^2\sigma$) and reducing the thermal conductivity[7–10].

Widespread application of thermoelectric technology calls for low-cost, environmentally benign and nontoxic high-performance thermoelectric materials. Current commercial materials contain a large amount of the rare element tellurium or the toxic element lead (for example, $Bi_2Te_3$ or PbTe) that prohibits large-scale applications. Zintl compounds such as skutterudites[11], clathrates[12] and zinc antimonides[13] are a promising class of thermoelectric materials with high figure of merit. However, these materials often contain expensive, scarce or toxic heavy elements such as Ge, Co, Yb, Eu and Cd. Among them, $Mg_3Sb_2$-based Zintl compounds are a potential source of environmentally friendly, earth-abundant and inexpensive materials. The earth abundance of magnesium is several orders of magnitude larger than that of other elements like Zn, Co, Ga and Ge in the earth's crust. $Mg_3Sb_2$ was initially investigated with the hope of realizing high $zT$ value for the intermediate- and high-temperature applications[14]. However, the thermoelectric performance of $Mg_3Sb_2$ is rather low and severely limited by poor electrical transport properties, even though it has reasonably low thermal conductivity. Many strategies, including theoretical orbital engineering[4] and experimental tuning of hole carrier concentration via various dopants[15–18], have been developed for optimizing p-type performance, whereas nearly no attempt[19] has been made for n-type doping in these compounds. In fact, all known Zintl antimonides except skutterudites have been reported to be p-type, and this appears to be because of intrinsic point defects[20]. It is a great challenge to achieve n-type properties in $Mg_3Sb_2$-based Zintl compounds.

Using combined theoretical prediction and experimental validation, here we present successful n-type doping in $Mg_3Sb_{1.5}Bi_{0.5}$ solid solution using tellurium and obtain a strongly enhanced $zT$ over a wide range of temperatures, from 0.56 to 1.65 at $300-725$ K in $Mg_3Sb_{1.48}Bi_{0.48}Te_{0.04}$, compared with p-type undoped $Mg_3Sb_2$ (ref. 16) with $zT$ from 0.002 to 0.26 at $332-750$ K (Fig. 1a). The thermoelectric performance of n-type $Mg_3Sb_{1.5-0.5x}Bi_{0.5-0.5x}Te_x$ ($x = 0.04$, 0.05 and 0.08) is at least 2 times higher than that of p-type $Mg_3Sb_2$-based compounds including Na-doped $Mg_3Sb_2$ (ref. 18) and $Mg_3Sb_{1.8}Bi_{0.2}$ (ref. 16) throughout the whole temperature range, outperforming other p-type $CaAl_2Si_2$-type Zintl compounds[21,22] including $EuZn_{1.8}Cd_{0.2}Sb_2$ and $YbCd_{1.6}Zn_{0.4}Sb_2$ that have the highest reported thermoelectric performance so far. Moreover, high n-type performance at $300-725$ K shown in $Mg_3Sb_{1.48}Bi_{0.48}Te_{0.04}$ is comparable to the best current state-of-the-art n-type materials[5,12,23–26] such as $Bi_2Te_{3-x}Se_x$ and $AgPb_mSbTe_{2+m}$ (Fig. 1b) that, however, contain a large amount of expensive, scarce or toxic heavy elements such as Te, Ag and Pb. Combining theory and experiment, we show that the exceptionally high thermoelectric performance originates from a considerably enhanced power factor that is aided by the combination of a low resistivity and an enhanced Seebeck coefficient contributed by the multiple band behaviour dominated by a unique near-edge conduction band with a high valley degeneracy of 6.

## Results

**Multi-valley conduction bands in n-type $Mg_3Sb_2$.** The $Mg_3Sb_2$ Zintl compound with a crystal structure similar to $CaAl_2Si_2$ is intrinsically p-type that might be attributed to the point defects at Mg sites[20]. Many efforts have been directed towards p-type doping using a variety of dopants, whereas nearly no attempt has been made for n-type doping. The present work is intuitively focussed on searching for potential high performance n-type thermoelectric candidates from Zintl compounds. By combining full band structure calculations and semiclassical Boltzmann transport theory, the Seebeck coefficient and power factor of both n-type and p-type $Mg_3Sb_2$ are estimated and shown in Fig. 2a,b. We find that the Seebeck coefficient and power factor of n-type doping are much better than those of p-type doping for binary $Mg_3Sb_2$. Interestingly, the n-type electrical transport properties are found to be related to a unique conduction band with 6 conducting carrier pockets along the M–L line (Fig. 2c,d).

Several methods shown in Fig. 2 are used to elucidate the multiple band characteristics in n-type $Mg_3Sb_2$, responsible for the exceptional electrical transport performance. *Ab initio* band structure calculation by density functional theory (DFT) shows an indirect band gap of 0.6 eV in $Mg_3Sb_2$ with a valence band maximum located at the $\Gamma$ point and a conduction band minimum at the K point (Fig. 2c). The conduction band at the K point (the K band) in the Brillouin zone possesses a valley degeneracy of 2. However, there is a secondary conduction band (the ML band) located just above the K band along the M–L high

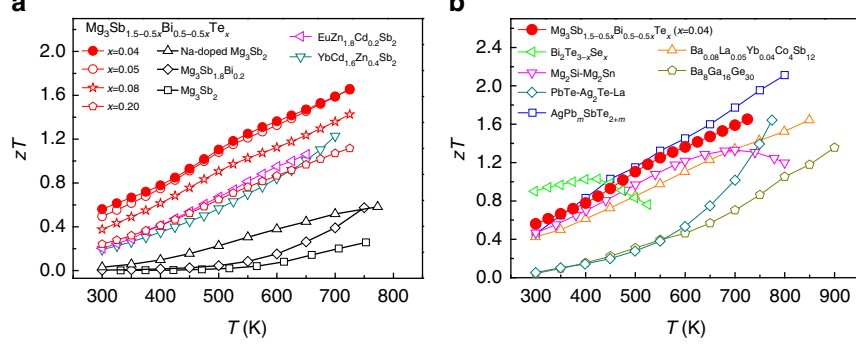

**Figure 1 | $zT$ value of n-type $Mg_3Sb_2$-based Zintl compound.** (**a**) $zT$ values of n-type $Mg_3Sb_{1.5-0.5x}Bi_{0.5-0.5x}Te_x$ ($x = 0.04$, 0.05, 0.08 and 0.20) in comparison with p-type undoped $Mg_3Sb_2$ (ref. 16), Na-doped $Mg_3Sb_2$ (ref. 18), $Mg_3Sb_{1.8}Bi_{0.2}$ (ref. 16), $EuZn_{1.8}Cd_{0.2}Sb_2$ (ref. 22) and $YbCd_{1.6}Zn_{0.4}Sb_2$ (ref. 21). (**b**) $zT$ comparison of n-type $Mg_3Sb_{1.48}Bi_{0.48}Te_{0.04}$ and current state-of-the-art n-type thermoelectric materials, Cu-doped $Bi_2Te_{3-x}Se_x$ (ref. 23), $AgPb_mSbTe_{2+m}$ (ref. 24), $Mg_2Si-Mg_2Sn$ (ref. 5), $Ba_{0.08}La_{0.05}Yb_{0.04}Co_4Sb_{12}$ (ref. 25), PbTe-$Ag_2$Te-La (ref. 26) and $Ba_8Ga_{16}Ge_{30}$ (ref. 12).

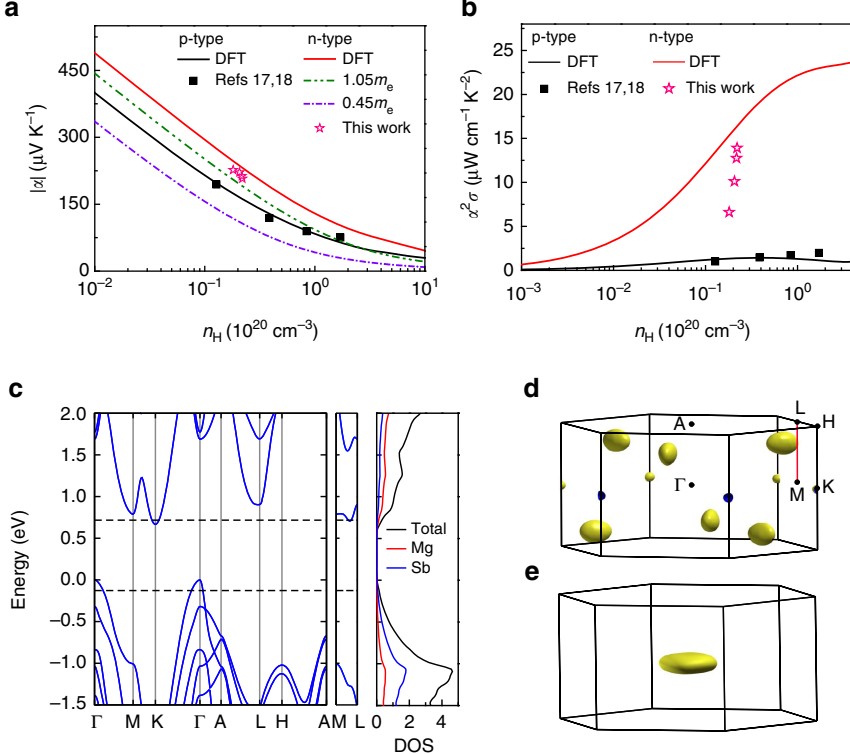

**Figure 2 | Strongly enhanced electrical transport properties induced by multi-valley conduction bands in n-type Mg$_3$Sb$_2$.** (**a**) The magnitude of Seebeck coefficient ($|\alpha|$) versus Hall carrier concentration ($n_H$) at 300 K. (**b**) Power factor as a function of Hall carrier concentration. In (**a**,**b**), the black and red solid lines represent the prediction of p-type and n-type Mg$_3$Sb$_2$ from full density functional theory (DFT) band structure calculation. Green and purple dashed lines show the expected $|\alpha|$ versus $n_H$ behaviour for single parabolic bands with effective masses equal to the two individual conduction bands ML and K at the band edges. The black solid points are the reported p-type doped Mg$_3$Sb$_2$ from the refs 17,18. The pink star points represent n-type Te-doped Mg$_3$Sb$_{1.5}$Bi$_{0.5}$ Zintl compounds from our own work. Our experimental data lie below the curve by DFT for Mg$_3$Sb$_2$ because there is an energy difference of 0.12 eV between the ML band and K band in n-type Mg$_3$Sb$_{1.5}$Bi$_{0.5}$, whereas these two bands are nearly converged in Mg$_3$Sb$_2$ (see Fig. 3a). (**c**) Calculated electronic band structure and partial density of states (DOS) for Mg$_3$Sb$_2$. (**d**,**e**) Calculated Fermi surfaces of (**d**) n-type and (**e**) p-type Mg$_3$Sb$_2$ at the Fermi level 0.03 eV above conduction band minimum and 0.1 eV below valence band maximum, respectively. The front sides of Fermi surfaces are plotted in yellow, whereas the back sides are coloured in blue. Black dots represent the high-symmetry k-points. The high-symmetry M-L line is marked in red colour. Fermi surface of n-type Mg$_3$Sb$_2$ exhibits 6 anisotropic carrier pockets along the M-L line and 6 one-third pockets at the K point, whereas p-type Mg$_3$Sb$_2$ possesses only one highly anisotropic carrier pocket at the $\Gamma$ point.

symmetry line and this ML band shows a valley degeneracy of 6. The calculation predicts a rather small energy difference of 0.02 eV between these two conduction bands. Such a small energy difference, comparable to $k_BT$ at room temperature, implies that the two conduction bands are nearly converged. The ML band covers an energy range of 0.19 eV at the band edge and in this energy window a steep increase of density of states (DOS) appears (Fig. 2c). Moreover, there are two other conduction bands (the M band and L band) located at 0.12 and 0.21 eV above the K band. In contrast to the multi-valley bands at the conduction band minimum, there is only one near-edge valence band at the $\Gamma$ point (the $\Gamma$ band). As a result of the large DOS induced by multiple conduction bands, the Fermi level slowly moves up the conduction bands with increasing n-type doping concentration. At room temperature, the ML band can be easily reached as the electron doping concentration approaches ~$4 \times 10^{19}$ cm$^{-3}$, whereas approaching the M or L band minimum will require a doping level as high as ~$2 \times 10^{20}$ or ~$6 \times 10^{20}$ cm$^{-3}$, respectively.

The multi-valley band feature can be directly observed by the iso-energy Fermi surface (see Fig. 2d). The iso-energy Fermi surface of n-type Mg$_3$Sb$_2$ for an energy level 0.03 eV above conduction band minimum shows 6 isolated full electron pockets along the M–L line inside the Brillouin zone and 6 one-third

pockets at the K point. Accordingly, the valley degeneracies of the ML band and the K band are 6 and 2, respectively, that add up to 8. Such a high valley degeneracy is comparable to many widely used thermoelectric materials such as (Bi, Sb)$_2$Te$_3$ (ref. 27). In contrast, the Fermi surface of p-type Mg$_3$Sb$_2$ only shows one hole pocket at the $\Gamma$ point (Fig. 2e). Given the above result, it is obvious that n-type doping will display much higher degeneracy of carrier pockets than that of p-type doping in Mg$_3$Sb$_2$.

Another clear illustration of the complex band behaviour can be seen in the doping-dependent Seebeck coefficient. The doping dependence of the Seebeck coefficient at 300 K shown in Fig. 2a is simulated using full band structure calculations as well as a single parabolic band model. The experimental data of p-type doped Mg$_3$Sb$_2$ agree very well with the curve simulated by integrated *ab initio* full band structure and semiclassical Boltzmann transport theory under a rigid band approximation, proving the effectiveness of this approach. The Seebeck coefficient is proportional to the DOS effective mass[27,28], given by $m_d^\star = N_v^{2/3} m_s^\star$, where $N_v$ represents the valley degeneracy and $m_s^\star$ is the single valley effective mass. The effective masses $m_s^\star$ of the K band and the ML band are respectively $0.28m_e$ and $0.32m_e$, indicating light mass behaviours. However, owing to high valley degeneracy ($N_v = 6$) the DOS effective mass of the ML band $m_d^\star = 1.05m_e$ is much heavier than that of the K band ($m_d^\star = 0.45m_e$, $N_v = 2$).

The Seebeck coefficient estimated by a single band model with DOS effective mass of $0.45m_e$ from the K band is only $-105\,\mu V\,K^{-1}$ at $2.2\times10^{19}\,cm^{-3}$, much smaller than full band structure calculated Seebeck coefficient of $-230\,\mu V\,K^{-1}$ at the same carrier concentration. Even using a single band model with the heavy mass from the ML band, the Seebeck coefficient ($-190\,\mu V\,K^{-1}$) at the same carrier concentration gives a value that is still lower than that of multiband DFT model. Therefore, a single parabolic band model cannot reproduce the Seebeck coefficient from full DFT band structure calculation, confirming the multiple band effects in n-type $Mg_3Sb_2$. Although the effective mass $m_s^\star=0.58m_e$ of the $\Gamma$ band at the valence band maximum is relatively heavier than those of the K band and ML band, the low valley degeneracy ($N_v=1$) leads to a DOS effective mass of $0.58m_e$ of the $\Gamma$ valence band, much smaller than that of the ML band. As expected, the Seebeck coefficient of p-type $Mg_3Sb_2$ predicted by full DFT calculation is lower than that of the single band model estimated by the ML band effective mass. The above results reveal that the ML band with a high valley degeneracy of 6 makes an important contribution to the high Seebeck values.

The optimal electrical transport performance is determined by the weighted mobility, $\mu(m_d^\star/m_e)^{3/2}$, where $\mu$ is the carrier mobility and $m_e$ is the mass of an electron[3,28]. Taking the assumption of acoustic phonon scattering for charge carriers, the carrier mobility can be expressed as $\mu\propto1/m_s^{\star5/2}$ (ref. 27). Considering the expression of the DOS effective mass, the weighted mobility can be simplified and expressed as proportional to $N_v/m_s^\star$. Thus, the ML band with a sixfold

valley degeneracy and light effective mass is highly desirable for the electrical performance. The high valley degeneracy $N_v=6$ has the effect of producing a large DOS effective mass and thereby a high Seebeck coefficient without explicitly reducing the mobility. In addition, compared with the $\Gamma$ valence band, the light effective mass $m_s^\star$ of the ML band is beneficial to increase $\mu$ and thus improve electrical conductivity $\sigma$. Therefore, it is clear that the ML band plays a crucial role in the strongly enhanced power factor shown in Fig. 2b.

**Exploring potential candidates with multi-valley conduction bands.** In order to understand whether the conduction band minimum along the M–L line is unique in $Mg_3Sb_2$, we conduct a screening of the band structures from a variety of $CaAl_2Si_2$-type Zintl compounds using our previous computational methods[4]. As a result, we find that the conduction band minimum along the M–L line only exists in binary $Mg_3X_2$ (X = As, Sb, Bi), shown in Fig. 2c and Supplementary Fig. 1. The band gaps decrease as X goes down the periodic table from As (1.6 eV) to Sb (0.6 eV) to Bi (semimetal). The conduction bands at the band edges of these compounds depict very similar multiple band behaviours at the K point and along the M–L line. This is because the near-edge conduction bands are dominated by the electronic states of Mg, whereas the valence bands are contributed by Sb (Fig. 2c). Hence, doping at anion sites will only have a minor effect on the conduction band minimum and thereby will not destroy the multi-valley conduction bands. Therefore, we can expect that

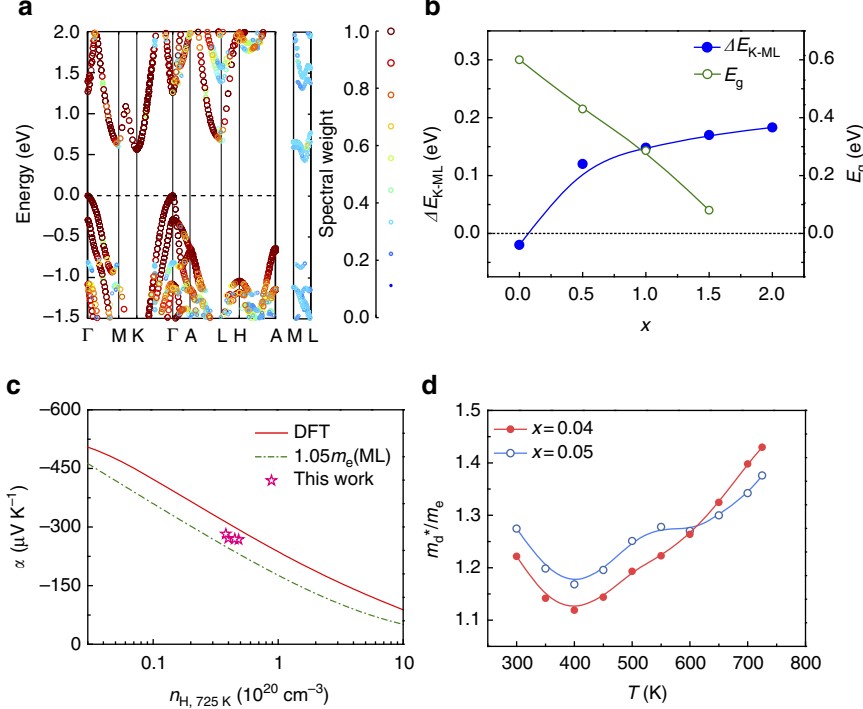

**Figure 3 | Multiple conduction band behaviour in n-type $Mg_3Sb_{2-x}Bi_x$ solid solutions.** (**a**) Effective band structure of $Mg_3Sb_{1.5}Bi_{0.5}$ solid solution. The spectral weight is represented by colour and the symbol size. Points with the spectral weight of $<0.25$ are filtered out. The band structure of $Mg_3Sb_{1.5}Bi_{0.5}$ depicts a multiple conduction band behaviour similar to that of $Mg_3Sb_2$ (Fig. 2c), where the ML band possesses a sixfold valley degeneracy and the K band has a twofold valley degeneracy. However, the ML band in $Mg_3Sb_{1.5}Bi_{0.5}$ becomes the conduction band minimum that is $\sim0.12\,eV$ below the K band. (**b**) The energy difference $\Delta E_{K-ML}$ between the K band and the ML band and the band gap $E_g$ as a function of the fraction $x$ in $Mg_3Sb_{2-x}Bi_x$ solid solutions. $Mg_3Bi_2$ ($x=2.0$) is a semimetal. The solid lines represent fitted curves using a B spline. (**c**) Seebeck coefficient versus Hall carrier concentration at 725 K. The red solid line represents the prediction of n-type $Mg_3Sb_2$ from full density functional theory (DFT) band structure calculation. The green dashed line depicts the expected $\alpha$ versus $n_H$ behaviour at 725 K for a single parabolic band with an effective mass equal to the ML conduction band. The pink star points are the data of Te-doped $Mg_3Sb_{1.5}Bi_{0.5}$ of this work. (**d**) Temperature dependence of the experimental DOS effective mass calculated from a single band model. The solid lines represent fitted curves using a B spline.

n-type doped $Mg_3Sb_2$ and $Mg_3As_2$ as well as solid solutions $Mg_3Sb_{2-x}Bi_x$, $Mg_3Sb_{2-x}As_x$ and $Mg_3As_{2-x}Bi_x$ with suitable band gaps will show enhanced power factor as well as thermoelectric performance if properly doped at the anion sites.

**Multiple conduction band behaviour in n-type $Mg_3Sb_{1.5}Bi_{0.5}$.** To confirm the existence of multiple conduction bands and especially the ML band in the solid solutions, the electronic structures of $Mg_3Sb_{2-x}Bi_x$ solid solutions are simulated by DFT and the band structure of an exemplified solid solution $Mg_3Sb_{1.5}Bi_{0.5}$ is shown in Fig. 3a. The result proves that multiple band behaviour including the ML band with a sixfold valley degeneracy is indeed preserved in $Mg_3Sb_{1.5}Bi_{0.5}$ solid solution and the dispersions and effective masses of the K band and ML band are very similar to those of $Mg_3Sb_2$. The main difference, however, is that the ML band in $Mg_3Sb_{1.5}Bi_{0.5}$ is shifted downward 0.12 eV below the K band and therefore becomes the conduction band minimum. This is good for the thermoelectric performance of $Mg_3Sb_{1.5}Bi_{0.5}$ as the ML band with a high valley degeneracy of 6 can be easily reached with a relatively low doping level. Thus, it is of interest to understand the correlation between the composition of $Mg_3Sb_{2-x}Bi_x$ solid solutions and the energy difference of the K band and ML band, defined as $\Delta E_{K-ML} = E_K - E_{ML}$. The dependence of $\Delta E_{K-ML}$ on the fraction $x$ is shown in Fig. 3b. Surprisingly, $\Delta E_{K-ML}$ can be continuously tuned by the fraction $x$ from $-0.02$ to $0.18$ eV. Energy gaps of $Mg_3Sb_{2-x}Bi_x$ show a decreasing trend from 0.43 to 0.08 eV as $x$ increases from 0.5 to 1.5 (Fig. 3b), suggesting that the bipolar effect will be obvious when $x > 1$. The obvious bipolar effect for $Mg_3Sb_{2-x}Bi_x$ ($x > 1$) is confirmed in the previous experimental report[17]. Hence, n-type $Mg_3Sb_{2-x}Bi_x$ ($x \leq 1$) compounds are very promising thermoelectric candidates if properly doped on the anion sites.

Achieving n-type $Mg_3Sb_2$ by doping tellurium on the anion sites has been attempted and found to be difficult, whereas it is easier in $Mg_3Sb_{2-x}Bi_x$ solid solutions. This is probably because of the lower formation energy of tellurium doping on the anion sites of $Mg_3Sb_{2-x}Bi_x$ solid solutions (see one example in Supplementary Fig. 2). From the above theoretical calculation, $Mg_3Sb_{1.5}Bi_{0.5}$ possesses a small band gap of 0.43 eV as well as $\Delta E_{K-ML} = 0.12$ eV with the ML band as the conduction band minimum (Fig. 3a,b), making it a potential candidate for n-type doping. The experimental validation is successfully demonstrated in n-type doped $Mg_3Sb_{1.5}Bi_{0.5}$ solid solution using tellurium as an effective dopant. $Mg_3Sb_{1.5-0.5x}Bi_{0.5-0.5x}Te_x$ ($x = 0.04$, 0.05, 0.08 and 0.20) samples were synthesized by combining arc-melting process and spark plasma sintering (Supplementary Figs 3–5). The room temperature carrier concentration data of these samples show a small distribution from $1.18 \times 10^{19}$ ($x = 0.20$) to $2.20 \times 10^{19}$ cm$^{-3}$ ($x = 0.04$) (see Supplementary Fig. 6a) and the corresponding doping levels are deep enough to cut the ML band.

The experimental Seebeck coefficients of n-type $Mg_3Sb_{1.5-0.5x}Bi_{0.5-0.5x}Te_x$ samples at 300 or 725 K are larger than the Seebeck value calculated by a single band model using the DOS effective mass of the ML band (see Figs 2a and 3c). This result not only confirms the theoretical calculation that the ML band dominates the conduction band minimum of $Mg_3Sb_{1.5}Bi_{0.5}$, but also reveals that the K band located at 0.12 eV above the conduction band minimum makes a contribution to both the room-temperature and high-temperature electrical transports. As there is an energy difference of 0.12 eV between the ML band and the K band in n-type $Mg_3Sb_{1.5}Bi_{0.5}$, the experimental Seebeck coefficients of n-type $Mg_3Sb_{1.5-0.5x}Bi_{0.5-0.5x}Te_x$ are smaller than the calculated value by DFT for n-type $Mg_3Sb_2$ with the effective convergence of the two bands (see Figs 2c and 3a,b).

The carrier concentration of $Mg_3Sb_{1.5-0.5x}Bi_{0.5-0.5x}Te_x$ increases with increasing temperature and reaches $4.83 \times 10^{19}$ cm$^{-3}$ at 725 K in the high-performance sample with $x = 0.04$ (Supplementary Fig. 6a), suggesting that the Fermi

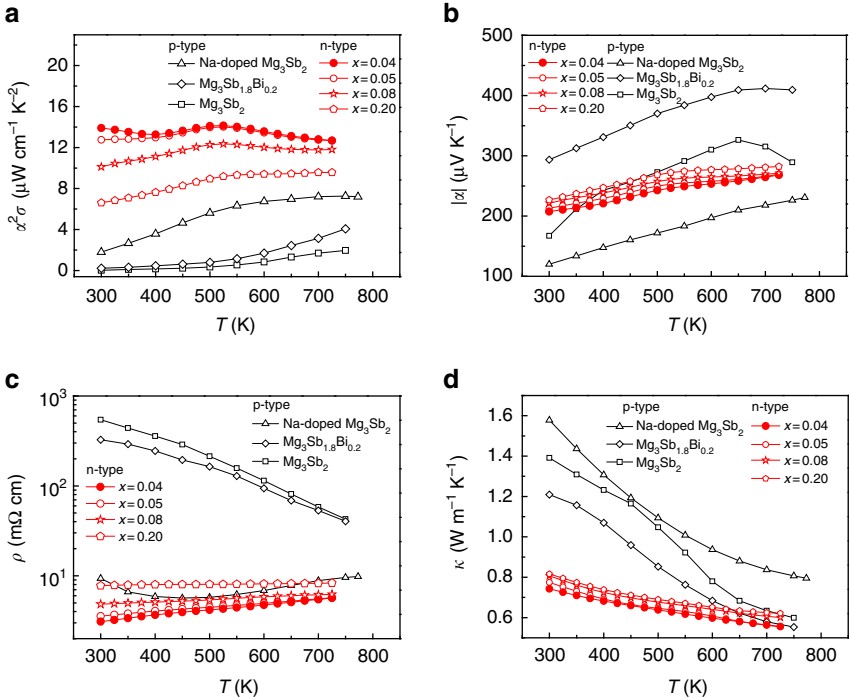

**Figure 4 | Thermoelectric transport properties of n-type $Mg_3Sb_{1.5-0.5x}Bi_{0.5-0.5x}Te_x$.** Temperature dependence of (**a**) power factor $\alpha^2\sigma$, (**b**) absolute values of Seebeck coefficient $\alpha$, (**c**) electrical resistivity $\rho$ and (**d**) total thermal conductivity $\kappa$ of $Mg_3Sb_{1.5-0.5x}Bi_{0.5-0.5x}Te_x$ ($x = 0.04$, 0.05, 0.08 and 0.20) and the comparison with p-type undoped $Mg_3Sb_2$ (ref. 16), $Na_{0.006}Mg_{2.994}Sb_2$ (ref. 18) and $Mg_3Sb_{1.8}Bi_{0.2}$ (ref. 16).

level will move upward approaching the K band with rising temperature. In addition, the broadening of the Fermi distribution makes the Fermi level easier to reach the K band at high temperatures. The temperature-dependent DOS effective masses in Te-doped $Mg_3Sb_{1.5}Bi_{0.5}$ are illustrated in Fig. 3d. As shown in Fig. 3d, the DOS effective mass of $Mg_3Sb_{1.5-0.5x}Bi_{0.5-0.5x}Te_x$ ($x = 0.04$ and $0.05$) derived from the experimental Seebeck coefficient increases with increasing temperature at 400–725 K, ruling out the single band behaviour at high temperatures. The above results again prove that the multiple band behaviour, including the effects from both the ML band and the K band, makes a contribution to the high-temperature transport properties.

**Thermoelectric properties of n-type Te-doped $Mg_3Sb_{1.5}Bi_{0.5}$.** Figure 4a shows the temperature-dependent power factors of n-type $Mg_3Sb_{1.5-0.5x}Bi_{0.5-0.5x}Te_x$ ($x = 0.04$, $0.05$, $0.08$ and $0.20$) samples. Among the n-type samples, $Mg_3Sb_{1.48}Bi_{0.48}Te_{0.04}$ has the highest power factor within the entire measurement temperature range. At room temperature, $Mg_3Sb_{1.48}Bi_{0.48}Te_{0.04}$ sample shows $\alpha^2\sigma = 13.9$ $\mu W\,cm^{-1}\,K^{-2}$ that reaches a maximum value of $14.14\,\mu W\,cm^{-1}\,K^{-2}$ at 525 K and then decreases. Compared with other state-of-the-art thermoelectric materials, the power factors of n-type $Mg_3Sb_{1.5-0.5x}Bi_{0.5-0.5x}Te_x$ samples are moderate, but much better than those of p-type $Mg_3Sb_2$-based materials ($Mg_3Sb_2$, Na-doped $Mg_3Sb_2$ and $Mg_3Sb_{1.8}Bi_{0.2}$). We attribute the significant enhancements of the power factors in Te-doped $Mg_3Sb_{1.5}Bi_{0.5}$ samples to the combination of the enhanced Seebeck coefficients and low resistivity values. The enhanced Seebeck coefficients are clearly demonstrated in the doping dependence of the Seebeck coefficients (Fig. 2a) as discussed above. In addition, the Seebeck coefficients of all Te-doped samples show increasing trends with increasing temperature. Typically, the $Mg_3Sb_{1.5-0.5x}Bi_{0.5-0.5x}Te_x$ sample with $x = 0.04$ exhibits the lowest absolute value of Seebeck coefficient that increases from $207\,\mu V\,K^{-1}$ at 300 K to $268\,\mu V\,K^{-1}$ at 725 K (Fig. 4b), higher than that of Na-doped $Mg_3Sb_2$ (ref. 18) at a comparable carrier concentration of $3.87 \times 10^{19}\,cm^{-3}$. The room temperature Seebeck coefficient is comparable to other state-of-the-art thermoelectric materials such as 1% Cu-doped $Bi_2Te_{3-x}Se_x$ (ref. 23) with $\alpha = -188\,\mu V\,K^{-1}$ at a similar carrier concentration.

The resistivity at room temperature shown in Fig. 4c decreases from $545\,m\Omega\,cm$ in undoped $Mg_3Sb_2$ to $3.09\,m\Omega\,cm$ in $Mg_3Sb_{1.48}Bi_{0.48}Te_{0.04}$, changing the temperature dependence from a semiconductor behaviour to a metallic behaviour. For the Te-doped samples, the resistivity $\rho$ increases with increasing Te content, showing a similar trend as the Seebeck coefficient. For the $Mg_3Sb_{1.48}Bi_{0.48}Te_{0.04}$ sample with the highest power factor, $\rho$ increases from $3.09\,m\Omega\,cm$ at 300 K to $5.67\,m\Omega\,cm$ at 725 K (Fig. 4c). The resistivity values of $Mg_3Sb_{1.5-0.5x}Bi_{0.5-0.5x}Te_x$ ($x = 0.04$, $0.05$ and $0.08$) are lower than that of Na-doped $Mg_3Sb_2$ (ref. 18) at a comparable carrier concentration ($3.87 \times 10^{19}\,cm^{-3}$). The low resistivity values of these samples at room temperature come from high mobility data of $62.4$–$91.8\,cm^2\,V^{-1}\,s^{-1}$ (Supplementary Fig. 6b), a factor of at least 3.7 larger than the mobility[17,18] of the undoped $Mg_3Sb_2$ ($\mu = 16\,cm^2\,V^{-1}\,s^{-1}$ at $n_H = 1.3 \times 10^{19}\,cm^{-3}$) and Na-doped $Mg_3Sb_2$ ($\mu = 16.7\,cm^2\,V^{-1}\,s^{-1}$ at $n_H = 3.87 \times 10^{19}\,cm^{-3}$). This reveals the light mass conduction band at the band edge induced by the ML band compared with the $\Gamma$ valence band, consistent with the theoretical calculation. In addition, the carrier mobility data of n-type Te-doped samples with $x = 0.04$, $0.05$ and $0.08$ roughly follow the temperature-dependent relation $\mu \propto T^{-P}$ ($1 \le P \le 1.5$) (Supplementary Fig. 6b), indicating that the

electrical transport is dominated by acoustic phonon scattering. Using the mobility of the undoped $Mg_3Sb_2$ and $m_s^\star$ of the ML band ($0.32m_e$) and the $\Gamma$ band ($0.58m_e$), the mobility of n-type $Mg_3Sb_{1.5}Bi_{0.5}$ can be roughly estimated by $\mu \propto 1/m_s^{\star 5/2}$ relation to be $\sim 71\,cm^2\,V^{-1}\,s^{-1}$, comparable to the experimental values of the samples with $x = 0.04$, $0.05$ and $0.08$. The above results thus confirm that the light mass of the ML band in Te-doped $Mg_3Sb_{1.5}Bi_{0.5}$ indeed results in high mobility and thereby low resistivity that is favourable to the electrical transport.

The total thermal conductivity values of n-type $Mg_3Sb_{1.5-0.5x}Bi_{0.5-0.5x}Te_x$ ($x = 0.04$, $0.05$, $0.08$ and $0.20$) samples are low and exhibit decreasing trends with increasing temperature (Fig. 4d). The lowest room-temperature thermal conductivity of $0.743\,W\,m^{-1}\,K^{-1}$ is observed for the sample with $x = 0.04$. The total thermal conductivity, $\kappa$, value decreases to $0.556\,W\,m^{-1}\,K^{-1}$ at 725 K. Compared with p-type $Mg_3Sb_2$ and $Mg_3Sb_{1.8}Bi_{0.2}$, the observed reduction in $\kappa$ of Te-doped $Mg_3Sb_{1.5}Bi_{0.5}$ is expected from the phonon scattering of alloys. It means that the thermal conductivity definitely also makes a significant contribution to high $zT$ at low temperatures. However, compared with the reduction of thermal conductivity, the improvement of power factor at low temperatures in n-type Te-doped $Mg_3Sb_{1.5}Bi_{0.5}$ is much larger than those of p-type $Mg_3Sb_2$ and $Mg_3Sb_{1.8}Bi_{0.2}$. Moreover, the thermal conductivity values of both p-type and n-type compounds at high temperatures are comparable, again confirming that the high thermoelectric performance of n-type Te-doped $Mg_3Sb_{1.5}Bi_{0.5}$ mainly originates from the enhanced power factor.

## Discussion

In summary, combining theory and experiment, we demonstrate in n-type $Mg_3Sb_2$-based Zintl compounds that multi-valley conduction band behaviour dominated by a light conduction band with 6 conducting carrier pockets leads to an enhanced Seebeck coefficient, a low resistivity and thereby a high power factor. Such a unique conduction band feature makes a key contribution to enhanced power factors and high $zT$ over a wide temperature range, enabling Te-doped $Mg_3Sb_{1.5}Bi_{0.5}$ to be a promising n-type thermoelectric candidate for low- and moderate-temperature applications. Our results thus provide an insightful guidance for the search for, and design of, high-performance n-type thermoelectric materials from Zintl compounds using multi-valley band engineering.

## Methods

**Sample synthesis.** The samples with nominal compositions $Mg_3Sb_{1.5-0.5x}Bi_{0.5-0.5x}Te_x$ ($x = 0.04$, $0.05$, $0.08$ and $0.20$) were synthesized by combining arc melting and spark plasma sintering (SPS) techniques. High-purity elements Sb pieces (99.9999%, Chempur), Bi pieces (99.999%, Chempur) and Te pieces (99.999%, Sigma Aldrich) were weighed, ground into powders ($< 100\,\mu m$) in an agate mortar and mixed in a ball mill mixer (SpectroMill, Chemplex Industries, Inc.) for 15 min. The mixed powders were then cold-pressed into pellets with the diameter of 12.7 mm. The pellets were completely melted by the arc melting process in an argon atmosphere using Edmund Bühler Mini Arc Melting System MAM-1GB. The arc melting process was repeated two times for both top and bottom sides of the pellets to obtain good homogeneity. The molten ingots were then ground into fine powders with particle sizes smaller than 63 $\mu m$ and then mixed with Mg powders (99.8%, $\le 44\,\mu m$, Alfa Aesar) in a ball mill mixer for 15 min. Approximately 2 g of the mixed powders were then loaded into a 12.7 mm diameter high-density graphite die protected by the graphite paper and sintered by SPS pressing under a pressure of 75 MPa. SPS sintering is conducted in vacuum by heating to 823 K in 11 min followed by a 2 min dwell, and then heating to 1123 K in 7 min and staying for another 4 min. SPS pressing was carried out using an SPS-515S instrument (SPS Syntex Inc., Japan).

**Structure characterization.** Powder X-ray diffraction measurements were carried out on the SPS-pressed pellets using a Rigaku Smartlab equipped with a Cu K$_\alpha$ source and parallel beam optic to check phase purity and lattice parameters (see Supplementary Note 1 and Supplementary Figs 3–5). Quantitative elemental analysis of the high-performance pellet with $x = 0.05$ was carried out on FEI Nova

Nano SEM 600 equipped with an element EDS X-ray detector. The result shown in Supplementary Table 1 was the average value from five randomly selected areas of the pellet.

**Thermoelectric transport property measurements.** The in-plane Hall coefficient ($R_H$) and resistivity $\rho$ were measured on the pellets using the Van der Pauw method in a magnetic field up to 1.25 T (ref. 29). Hall carrier concentration ($n_H$) was calculated by $1/eR_H$, where $e$ is the elementary charge. The Hall carrier mobility $\mu_H$ was then calculated using the relation $\mu_H = R_H/\rho$. The pellets were annealed by running the measurement during both heating and cooling for more than 3 cycles. After the first cycle, the Hall measurements of the pellets are consistent upon several repeated heating and cooling measurements. The resistivity and Hall data of the final cycle was used for the analysis. The Seebeck coefficients of the pellets were then measured from the slope of the thermopower versus temperature gradient using chromel-niobium thermocouples on an in-house system, similar to the one reported by Iwanaga et al.[30] The thermal diffusivity ($D$) from 300 to 725 K was measured using the laser flash method (Netzsch, LFA457) (see Supplementary Fig. 7). Heat capacity ($C_P$) was estimated using the Dulong–Petit law $C_P = 3k_B$ per atom for the temperature range from 300 to 725 K. The density ($d$) was measured by Archimedes method. Thermal conductivity was then calculated by $\kappa = dDC_P$. For a comparison, one high-performance pellet $Mg_3Sb_{1.5-0.5x}Bi_{0.5-0.5x}Te_x$ ($x = 0.05$) was polished and cut into a $2 \times 2 \times 9$ mm bar for the measurement of electrical transport properties, including electrical resistivity ($\rho$) and Seebeck coefficient ($\alpha$), using a ZEM-3 (ULVAC) apparatus under a helium atmosphere from 300 to 725 K (Supplementary Fig. 8 and Supplementary Note 2). The thermoelectric $zT$, obtained on the high-performance sample with $x = 0.05$ by the home-built system and ZEM-3 setup, shows good consistency between each other (Supplementary Fig. 9). The estimated measurement uncertainties are listed as follows: 5% for electrical resistivity, 5% for Seebeck coefficient and 7% for thermal diffusivity; the combined uncertainty for $zT$ is $\sim 20\%$.

**DFT calculations.** DFT calculations were carried out using a full-potential linear augmented plane-wave plus local orbitals method as implemented in the Wien2k code[31]. The relaxed structure parameters of $Mg_3X_2$ (X = As, Sb, Bi) from our previous work[4] were adopted. The electronic structure calculations were conducted using the TB-mBJ potential[32] to get accurate band gaps and spin–orbit coupling was included in the calculations. The plane wave cutoff parameter $R_{MT}K_{max}$ was set to 9 and the corresponding Brillouin zone was sampled by a $36 \times 36 \times 24$ k mesh. An energy convergence criterion of $10^{-4}$ eV was used. The Fermi surface was plotted with the program Xcrysden[33]. For the anisotropic ML conduction band, the effective masses were calculated to be $m_{xx} = 0.55m_e$, $m_{yy}^* = 0.21m_e$ and $m_{zz}^* = 0.28m_e$ (see Supplementary Note 3 for details). The conduction band at the K point was nearly isotropic with $m_{xx}^* = m_{yy}^* = 0.32m_e$ and $m_{zz}^* = 0.21m_e$. For the highly anisotropic $\Gamma$ valence band, $m_{xx}^* = m_{yy}^* = 1.15m_e$ and $m_{zz}^* = 0.15m_e$. Electrical transport property calculations of $Mg_3Sb_2$ (the DFT curves in Figs 2a,b and 3c) were carried out by combining the *ab initio* band structure calculations and the Boltzmann transport theory under the constant carrier scattering time approximation as implemented in the BoltzTraP code[34] (Supplementary Note 3 and Supplementary Fig. 10). To calculate the power factor curves of $Mg_3Sb_2$ shown in Fig. 2b, we need to estimate the carrier scattering time. Calculation details of the constant carrier scattering time $\tau$ are provided in Supplementary Note 4. For the discussion on Fig. 3d, the band structure was assumed to be rigid and independent of the temperature.

Band structure calculations of $Mg_3Sb_{2-x}Bi_x$ ($x = 0.5$, 1 and 1.5) solid solutions were carried out in supercells with 40 atoms ($2 \times 2 \times 2$ unit cell). Crystal structures including lattice constants and ionic positions were fully relaxed using the PBE functional[35] in the Vienna *ab initio* simulation package (VASP)[36]. The plane-wave energy cutoff was set at 400 eV. The energy and Hellmann–Feynman force convergence criteria were $10^{-4}$ eV and 0.01 eV Å$^{-1}$, respectively. A $6 \times 6 \times 4$ Monkhorst–Pack k mesh was used for crystal structure optimization. In $Mg_3Sb_{2-x}Bi_x$ solid solutions, there are 16 equivalent positions for Sb/Bi atoms, and thus the possible atomic concentrations of Bi are multiples of 1/16. The crystal structure of the supercell with the lowest energy was used for further calculations and analysis. Electronic structures of solid solutions including spin–orbit coupling effect were calculated by TB-mBJ potential[32] in Wien2k[31] code. The zone samplings were done with uniform $15 \times 15 \times 8$ k mesh grids. Effective band structures of $Mg_3Sb_{2-x}Bi_x$ were calculated by unfolding the band structures of supercells into the primitive cells using an effective band unfolding technique[37,38]. Calculation details of the band unfolding and defect formation energy are shown in the Supplementary Notes 5 and 6.

**Data availability.** The data that support these findings are available from the corresponding author on request.

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

## Acknowledgements

This work was supported by the Danish National Research Foundation (DNRF93) and the Danish Center for Scientific Computing. We thank N. Pryds of Technical University of Denmark for the use of ZEM-3 setup. We thank K.A. Borup, K.F.F. Fischer, H. Reardon and A.B. Blichfeld for discussions.

## Author contributions

J.Z. and B.B.I. designed the study. J.Z., L.S. and S.H.P synthesized samples, characterized structures and measured high-temperature thermoelectric properties. J.Z. performed theoretical calculations. H.Y. helped in some thermal transport measurements and provided discussions. L.T.H. helped in the electrical transport measurements. J.Z., L.S. and B.B.I. analyzed data. J.Z. and B.B.I. wrote the manuscript. All other authors read and edited the manuscript.

## Additional information

**Competing financial interests:** The authors declare no competing financial interests.

**Publisher's note**: 

