## [Peer Review File · Nature Communications]

Reviewers' comments:

Reviewer #1 (Remarks to the Author):

A. The authors report on the discovery of a new thermoelectric material (not 'materials' as it is mentioned in the title) which contains mainly magnesium and antimony together with bismuth and is additionally doped with tellurium. This is a well done study of an indeed interesting thermoelectric material.

B. The basic material is new but not novel, this kind of materials are in focus of the research groups at least since the 22 ICT (cf. DOI: 10.1109/ICT.2003.1287510), thermoelectric properties of Bi-substituted single crystals of Mg_3Sb_2 were recently reported by S. Kim et al. (2015) Mater. Res. Express 2 055903. The novelty of the manuscript is in the application of Te-doping yielding the n-type material.

C. The authors use techniques and methodologies well accepted in thermoelectric research.

D. Statistical approach does not apply. The study was made on a single sample.

E. Concerning the material composition reported in the manuscript I do not have doubts in the reliability of the results. The robustness and validity of the results can not be judged in general and should be considered with the usual criteria for the thermoelectric measurements made on the single specimen.

F. The report is written quite fragmentarily and is focused only on one specimen. In order to give a more complete picture of the results the following modifications of the manuscript are reasonable:

- Justification why the composition with $\text{Bi}_{0.5}$ was used, what properties have other compositions of the solid solution between Mg_3Sb_2 and Mg_3Bi_2 .

- Justification of the doping: why tellurium and not selenium or sulfur, why is 0.05 an optimum of the doping if it is).

- The multi-valley aspect of the band structure (and the whole calculational part of the work) is not really explained and looks overestimated. There is not clear from the manuscript how from the spectral-weight representation of the band structure single curve-representation for the Seebeck coefficient was obtained. The conclusion which can be drawn from figure 3 are not clear for the reader.

G. Cf. B.

H. In the abstract, the experimental part is suppressed, the calculational aspect is overestimated and presented in the form of the finally proven facts. In the manuscript, the 'six-fold valley degeneracy' is not even explicitly mentioned, e.g. in the comments to the figure 3.

The same impression makes also the summary. Here, the multi-valley approach is presented as a basis for the materials design, which was not the issue in the presented manuscript.

Reviewer #2 (Remarks to the Author):

This manuscript provides new insight into Mg_3Sb_2 structure and shows that the n-type doped material, Te doped $\text{Mg}_3\text{Sb}_{1.5}\text{Bi}_{0.5}$. The authors show that electronic calculations guided their decision to pursue the n-type material. This material, Te doped $\text{Mg}_3\text{Sb}_{1.5}\text{Bi}_{0.5}$ shows excellent thermoelectric properties with a high figure of merit, zT , ranging from 0.5 to 1.62 at 300 to 725 K, respectively. Based on their theoretical calculations, they propose that the thermoelectric performance is a result mainly of the power factor due to a near-edge conduction band with six-fold valley degeneracy. The guidance of theory is very important for this manuscript and provides the significance for the research because of several reasons. The first is that many of these Zintl phases are typically p-type and to date, there has been little exploration of n-type doping. In cases, where it has been attempted, there has been little success. The second reason is that understanding why a certain composition might have good properties is extremely important for the field. Finally, since high zT in these new Zintl phases has only recently been discovered, it points to significant improvements in the future,

highlighting the importance of Zintl phases for thermoelectric applications.

In the experimental, the source of the elements should be specified. The only other suggestion would have been to prepare a few more samples as I typically like to see a progression to high zT rather than just one sample. However, I assume that the authors are in the process of preparing more samples for a more complete paper and a communication is warranted with the combination of the theoretical support.

Reviewer #3 (Remarks to the Author):

It reports high thermoelectric performance in n-type Mg_3Sb_2 -based compound, which is achieved by Te doping in $Mg_3Sb_{1.5}Bi_{0.5}$. The authors explain the high Seebeck coefficient by multi-valley conduction band and the low electric resistivity by light conduction band, leading to high power factor. I do not recommend it to be published in Nat. Comm. with the following serious problems.

Major parts:

1. There are a variety of zT values depending on the method of sample preparation. In this sense, the authors need to show their own data on pristine Bi_2Sb_2 and $Bi_2Sb_{1.5}Bi_{0.5}$ samples, which need to be compared with their own theoretical results for p-type cases.

2. The authors only show the data of $Mg_3Sb_{1.5-0.5x}Bi_{0.5-0.5x}Te_x$ ($x=0.05$). Even though I do agree that n-type doping is effective using tellurium, it is more persuasive to show the data of $Mg_3Sb_{1.5}Bi_{0.5}$ with various Te dopants, x . It gives more convincing n_H dependence, for example more data points in Figs. 2a, 2b, and 3d.

3. The authors mentioned that n-type doping in Sb sites is a lot easier in $Mg_3Sb_{1.5}Bi_{0.5}$ solid solutions. This opinion needs to be demonstrated either experimentally (maybe, based on TEM or XPS?) or theoretically (maybe, based on formation energy or electronegativity?). Moreover, I doubt the nominal composition $Mg_3Sb_{1.5-0.5x}Bi_{0.5-0.5x}Te_x$ ($x=0.05$). According to their opinion, $Mg_3Sb_{1.5}Bi_{0.5-x}Te_x$ is more effective?

4. I believe that the authors did not consider temperature effect in their DFT calculations. Nevertheless, they compare the theoretical data of n-type Mg_3Sb_2 with the experimental data taken at 300 K (see Figs. 2a and 2b), and the theoretical data of n-type $Mg_3Sb_{1.5}Bi_{0.5}$ with the experimental data taken at 725 K (see Fig. 3d). If one considers different temperature, for example, 300 K in Fig. 3d and 725 K in Figs. 2a and 2b, the experimental data do not agree with the simulated curves.

Minor parts:

5. According to their n_H and μ_H data; when simple looking, n_H is almost 2 times increased with temperature and μ_H is almost 3 times decreased with temperature, the resistivity should be increased about 1.5 times, which seems to be inconsistent with Fig. 4c.

6. Also, the Seebeck coefficient at 725 K in Fig. 4b seems to be about 280 $\mu V/K$, but the data point in Fig. 3d is about 300 $\mu V/K$. They are not consistent each other.

7. The authors need to explain the reason on thermal hysteresis of Seebeck coefficient and electrical resistivity (Figs. S7a and S7b). If it originates from annealing effect and/or slight oxidation, the hysteresis may not be reversible.

Thank you for the three referee reports. We are grateful for their constructive and helpful comments, and below we address the reports point by point.

Reviewer 1

Comment 1

'D. Statistic approach does not apply. The study was made on single sample.'

E. Concerning the material composition reported in the manuscript I do not have doubts in the reliability of the results. The robustness and validity of the results cannot be judged in general and should be considered with the usual criterions for the thermoelectric measurements made on the single specimen.

F. The report is written quiet fragmentarily and is focused only on one specimen.'

Reply

We thank the referee for this comment. We agree that in general the robustness and validity of the results cannot be judged on a single sample. To address this concern, we have made more samples with nominal compositions $\text{Mg}_3\text{Sb}_{1.5-0.5x}\text{Bi}_{0.5-0.5x}\text{Te}_x$ ($x = 0.04, 0.05, 0.08, \text{ and } 0.20$), which have been characterized on our homebuilt systems. The high zT in the high-performance sample with $x = 0.05$, is measured both on the commercial ZEM-3 setup and the homebuilt system, and the two sets of data show very good agreement with each other (see Supplementary Fig. 9).

The manuscript has been rewritten to include the additional samples with more compositions. The major revision is summarized below:

- (a) Figures 1-4, Supplementary Figures 3-7, and their captions have been modified to include different compositions $\text{Mg}_3\text{Sb}_{1.5-0.5x}\text{Bi}_{0.5-0.5x}\text{Te}_x$ ($x = 0.04, 0.05, 0.08, \text{ and } 0.20$).
- (b) One sentence is modified in line 5 of the second paragraph on page 3: “The TE performance of n-type $\text{Mg}_3\text{Sb}_{1.5-0.5x}\text{Bi}_{0.5-0.5x}\text{Te}_x$ ($x = 0.04, 0.05, \text{ and } 0.08$) is at least 2 times higher than that of p-type Mg_3Sb_2 -based compounds including Na-doped Mg_3Sb_2 (ref. 18) and $\text{Mg}_3\text{Sb}_{1.8}\text{Bi}_{0.2}$ (ref. 16) throughout the whole temperature range”
- (c) Several sentences have been modified to include different compositions in the end of the second paragraph on page 9: “ $\text{Mg}_3\text{Sb}_{1.5-0.5x}\text{Bi}_{0.5-0.5x}\text{Te}_x$ ($x = 0.04, 0.05, 0.08, \text{ and } 0.20$) samples were synthesized by combining arc-melting process and spark plasma sintering (Supplementary Figs 3-5). The room-temperature carrier concentration data of these samples show a small distribution from 1.18×10^{19} ($x = 0.20$) to $2.20 \times 10^{19} \text{ cm}^{-3}$ ($x = 0.04$) (see Supplementary Fig. 6a) and the corresponding doping levels are deep enough to cut the ML band.”
- (d) In the end of page 10 several sentences have been modified as: “Figure 4a shows the temperature-dependent power factors of n-type $\text{Mg}_3\text{Sb}_{1.5-0.5x}\text{Bi}_{0.5-0.5x}\text{Te}_x$ ($x = 0.04, 0.05, 0.08, \text{ and } 0.20$) samples. Among the n-type samples, $\text{Mg}_3\text{Sb}_{1.48}\text{Bi}_{0.48}\text{Te}_{0.04}$ has the highest power factor within the entire measurement temperature range. At room temperature, $\text{Mg}_3\text{Sb}_{1.48}\text{Bi}_{0.48}\text{Te}_{0.04}$ sample shows $\alpha^2\sigma = 13.9 \mu\text{W cm}^{-1} \text{ K}^{-2}$, which reaches a maximum value of $\sim 14.14 \mu\text{W cm}^{-1} \text{ K}^{-2}$ at 525 K and then decreases.”
- (e) In the end of page 11 several sentences have been modified as: “For the Te-doped samples, the resistivity ρ increases with increasing Te content, showing a similar trend as the Seebeck coefficient. For the $\text{Mg}_3\text{Sb}_{1.48}\text{Bi}_{0.48}\text{Te}_{0.04}$ sample with the highest power factor, ρ increases from 3.09 m Ω cm at 300 K to 5.67 m Ω cm at 725 K (Fig. 4c). The resistivity values of $\text{Mg}_3\text{Sb}_{1.5-0.5x}\text{Bi}_{0.5-0.5x}\text{Te}_x$ ($x = 0.04, 0.05, \text{ and } 0.08$) are lower than that of Na-doped Mg_3Sb_2 (ref. 18) at a comparable carrier concentration ($3.87 \times 10^{19} \text{ cm}^{-3}$). The low resistivity values of these samples at room temperature come from high mobility data of $\sim 62.4\text{-}91.8 \text{ cm}^2 \text{ V}^{-1} \text{ s}^{-1}$ (Supplementary Fig. 6b), a factor of at least 3.7 larger than the mobility^{17,18} of the undoped Mg_3Sb_2 ”.
- (f) In the beginning of the second paragraph several sentences have been modified as: “The total thermal conductivity values of n-type $\text{Mg}_3\text{Sb}_{1.5-0.5x}\text{Bi}_{0.5-0.5x}\text{Te}_x$ ($x = 0.04, 0.05, 0.08, \text{ and } 0.20$) samples are low

and exhibit decreasing trends with increasing temperature (Fig. 4d). The lowest room-temperature thermal conductivity of $0.743 \text{ W m}^{-1} \text{ K}^{-1}$ is observed for the sample with $x = 0.04$. The total thermal conductivity, κ , value decreases to $\sim 0.556 \text{ W m}^{-1} \text{ K}^{-1}$ at 725 K.”

- (g) In sample synthesis of *Methods* on page 13: “The samples with nominal compositions $\text{Mg}_3\text{Sb}_{1.5-0.5x}\text{Bi}_{0.5-0.5x}\text{Te}_x$ ($x = 0.04, 0.05, 0.08, \text{ and } 0.20$) were synthesized by combining arc melting and SPS techniques.”
- (h) In the first paragraph of page 15 one sentence is added: “The Seebeck coefficients of the pellets were then measured from the slope of the thermopower versus temperature gradient using chromel-niobium thermocouples on an in-house system, which is similar to the one reported by Iwanaga et al³⁰.” In addition, one reference is added: “30. Iwanaga, S., Toberer, E. S., LaLonde, A. & Snyder, G. J. A high temperature apparatus for measurement of the Seebeck coefficient. *Rev. Sci. Instrum.* **82**, 063905 (2011).”
- (i) One sentence is added in line 13 of page 15: “The thermoelectric zT , obtained on the high-performance sample with $x = 0.05$ by the home-built system and ZEM-3 setup, shows good consistency between each other (Supplementary Fig. 9).”

Comment 2

‘In order to give a more complete picture of the results the following modifications of the manuscript are reasonable:

Justification why the composition with Bi0.5 was used, what properties have other compositions of the solid solution between Mg_3Sb_2 and Mg_3Bi_2 .’

Reply

Thank you for the comment. From our theoretical calculation, we have shown that the bipolar effect will be obvious when $x > 1$ in $\text{Mg}_3\text{Sb}_{2-x}\text{Bi}_x$. This is confirmed in the previous experimental report (*J. Electron. Mater.* 2013, 42, 1307-1312). Hence, n-type $\text{Mg}_3\text{Sb}_{2-x}\text{Bi}_x$ ($x \leq 1$) compounds are very promising TE candidates if properly doped on the anion sites. The composition with $\text{Bi}_{0.5}$ was chosen as a proof-of-concept, because $\text{Mg}_3\text{Sb}_{1.5}\text{Bi}_{0.5}$ possesses a suitable band gap of 0.43 eV as well as $\Delta E_{\text{K-ML}} = 0.12$ eV with the ML band as the conduction band minimum. This is good for the electrical transport since the ML band with a high valley degeneracy of 6 can be reached with a low doping concentration. In addition, the formation energy of Te doping on the anion sites was calculated and

shown in Supplementary Fig. 2, which indicates that Te doping on the anion sites of $\text{Mg}_3\text{Sb}_{1.5}\text{Bi}_{0.5}$ is easier than that of Mg_3Sb_2 .

We agree that other compositions with $x < 1$ and $x \neq 0.5$ might also show good performance. However, as a proof-of-concept paper, we have shown a progression to high zT in n-type $\text{Mg}_3\text{Sb}_{1.5}\text{Bi}_{0.5}$ samples with different Te compositions. It is an excellent idea to carry out a study of the full range of Bi compositions in a future study. Thanks.

- (a) The energy gap E_g versus x plot was added in Figure 3b.
- (b) Two sentences are added in the beginning of page 9: “The obvious bipolar effect for $\text{Mg}_3\text{Sb}_{2-x}\text{Bi}_x$ ($x > 1$) is confirmed in a previous experimental report¹⁷. Hence, n-type $\text{Mg}_3\text{Sb}_{2-x}\text{Bi}_x$ ($x \leq 1$) compounds are very promising TE candidates if properly doped on the anion sites.”
- (c) Two sentences have been modified in the second paragraph of page 9: “Achieving n-type Mg_3Sb_2 by doping tellurium on the anion sites has been attempted and found to be difficult, while it is easier in $\text{Mg}_3\text{Sb}_{2-x}\text{Bi}_x$ solid solutions. This is probably due to the lower formation energy of tellurium doping on the anion sites of $\text{Mg}_3\text{Sb}_{2-x}\text{Bi}_x$ solid solutions (see one example in Supplementary Fig. 2). From the above theoretical calculation, $\text{Mg}_3\text{Sb}_{1.5}\text{Bi}_{0.5}$ possesses a small band gap of 0.43 eV as well as $\Delta E_{K-ML} = 0.12$ eV with the ML band as the conduction band minimum (Fig. 3a), making it a potential candidate for n-type doping. The experimental validation is successfully demonstrated in n-type doped $\text{Mg}_3\text{Sb}_{1.5}\text{Bi}_{0.5}$ solid solution using tellurium as an effective dopant.”

Comment 3

‘Justification of the doping: why tellurium and not selenium or sulfur, why is 0.05 an optimum of the doping if it is).’

Reply

In this work we choose tellurium for n-type doping, because tellurium is the close neighbor of antimony in the Periodic Table. It means that tellurium has a similar ion radius and electronegativity relative to antimony and bismuth, which will possibly make it easier for tellurium to be doped on the Sb or Bi sites. In addition, tellurium has been widely used as an effective n-type dopant in thermoelectric research. We agree that selenium or sulfur doping might also be very interesting due to the abundance of Se and S elements and we believe that it could be a nice future work. Thanks for this comment.

To address the second part of the comment, we have made more samples with nominal compositions $\text{Mg}_3\text{Sb}_{1.5-0.5x}\text{Bi}_{0.5-0.5x}\text{Te}_x$ ($x = 0.04, 0.05, 0.08,$ and 0.20). As shown in the updated Fig. 1a, $x = 0.04$ is an optimum of Te doping, and the sample with $x = 0.05$ shows a comparable performance. Please see the reply to comment 1 for details.

Comment 4

'The multi-valley aspect of the band structure (and the whole calculational part of the work) is not really explained and looks overestimated. There is not clear from the manuscript how from the spectral-weight representation of the band structure single curve-representation for the Seebeck coefficient was obtained. The conclusion which can be drawn from figure 3 are not clear for the reader.'

Reply

We are sorry for the confusion about this point. In Figures 3 and 2a,b, the DFT model is for n-type Mg_3Sb_2 , which has the ML band and the K band converged. But our experimental data is on Te-doped $\text{Mg}_3\text{Sb}_{1.5}\text{Bi}_{0.5}$ which has an energy difference of 0.12 eV between the ML band and K band. This is why the DFT curve looks overestimated.

Unfortunately, it is impossible for semi-classical Boltzman theory to calculate Seebeck coefficient from the spectral-weight representation of the band structure of $\text{Mg}_3\text{Sb}_{1.5}\text{Bi}_{0.5}$. The spectral-weight representation of the band structure of $\text{Mg}_3\text{Sb}_{1.5}\text{Bi}_{0.5}$ was used to understand that the ML band moves downward 0.12 eV below the K band and becomes the conduction band minimum. Since the ML band with a high valley degeneracy of 6 is highly desirable for electrical transport, moving the ML band to the conduction band minimum is good and it can be reached with a low doping level.

In order to make the multiple band behavior of $\text{Mg}_3\text{Sb}_{1.5}\text{Bi}_{0.5}$ clear, the following revision has been applied:

- (a) The second paragraph of page 8: "The results prove that multiple band behavior including the ML band **with a six-fold valley degeneracy** is indeed preserved in $\text{Mg}_3\text{Sb}_{1.5}\text{Bi}_{0.5}$ solid solution and the dispersions and effective masses of the K band and ML band are very similar **to** those of Mg_3Sb_2 . The main difference, however, is that the ML band in $\text{Mg}_3\text{Sb}_{1.5}\text{Bi}_{0.5}$ is shifted downward **0.12 eV** below the K band and therefore becomes the conduction band minimum. This is good for the TE performance of $\text{Mg}_3\text{Sb}_{1.5}\text{Bi}_{0.5}$ since the ML band **with a high val-**

ley degeneracy of 6 can be easily reached with a relatively low doping level.”

- (b) The reason of DFT overestimation is explained in the third paragraph of page 9: “The experimental Seebeck coefficients of n-type $\text{Mg}_3\text{Sb}_{1.5-0.5x}\text{Bi}_{0.5-0.5x}\text{Te}_x$ samples at 300 K or 725 K are larger than the Seebeck value calculated by a single band model using the DOS effective mass of the ML band (See Figs 2a and 3c). This result not only confirms the theoretical calculation that the ML band dominates the conduction band minimum of $\text{Mg}_3\text{Sb}_{1.5}\text{Bi}_{0.5}$, but also reveals that the K band located 0.12 eV above the conduction band minimum makes a contribution to both the room-temperature and high-temperature electrical transports. Since there is an energy difference of 0.12 eV between the ML band and the K band in n-type $\text{Mg}_3\text{Sb}_{1.5}\text{Bi}_{0.5}$, the experimental Seebeck coefficients of n-type $\text{Mg}_3\text{Sb}_{1.5-0.5x}\text{Bi}_{0.5-0.5x}\text{Te}_x$ are smaller than the calculated value by DFT for n-type Mg_3Sb_2 with the effective convergence of the two bands (see Figs 3a,b and 2c).”
- (c) The second paragraph of page 10: “The carrier concentration of $\text{Mg}_3\text{Sb}_{1.5-0.5x}\text{Bi}_{0.5-0.5x}\text{Te}_x$ increases with increasing temperature and reaches $4.83 \times 10^{19} \text{ cm}^{-3}$ at 725 K in the high-performance sample with $x = 0.04$ (Supplementary Fig. 6a), suggesting that the Fermi level will move upward approaching the K band with rising temperature. Additionally, the broadening of the Fermi distribution makes the Fermi level easier to reach the K band at high temperatures. The temperature-dependent DOS effective mass in Te-doped $\text{Mg}_3\text{Sb}_{1.5}\text{Bi}_{0.5}$ are illustrated in Fig. 3d. As shown in Fig. 3d, the DOS effective mass of $\text{Mg}_3\text{Sb}_{1.5-0.5x}\text{Bi}_{0.5-0.5x}\text{Te}_x$ ($x=0.04$ and 0.05) derived from the experimental Seebeck coefficient increases with increasing temperature at 400-725 K, ruling out the single band behavior at high temperatures. The above results again prove that the multiple band behavior, including the effects from both the ML band and the K band, makes a contribution to the high temperature transport properties.”
- (d) One sentence is added to the caption of Figure 2 to explain the overestimation of DFT modelling: “Our experimental data lie below the curve by DFT for Mg_3Sb_2 , which is because there is an energy difference of 0.12 eV between the ML band and K band in n-type $\text{Mg}_3\text{Sb}_{1.5}\text{Bi}_{0.5}$ while these two bands are nearly converged in Mg_3Sb_2 (see Fig. 3a).”
- (e) Two sentences are added to the caption of Figure 3: “The band structure of $\text{Mg}_3\text{Sb}_{1.5}\text{Bi}_{0.5}$ depicts a multiple band behavior similar to that of Mg_3Sb_2 (Fig. 2c), where the ML band possesses a six-fold valley degeneracy and the K band has a two-fold valley degeneracy. However,

the ML band in $\text{Mg}_3\text{Sb}_{1.5}\text{Bi}_{0.5}$ becomes the conduction band minimum, which is about 0.12 eV below the K band.”

- (f) To the first paragraph of page 16, a few key words are added to make it clear: “Electrical transport property calculations of Mg_3Sb_2 (the DFT curves in Figs 2a,b and 3c) were carried out by combining the *ab initio* band structure calculations and the Boltzmann transport theory under the constant carrier scattering time approximation as implemented in the BoltzTraP code³⁴ (Supplementary Note 3 and Supplementary Fig. 10). To calculate the power factor curves of Mg_3Sb_2 shown in Fig. 2b, we need to estimate the carrier scattering time. Calculation details of the constant carrier scattering time τ are provided in Supplementary Note 4.”

Comment 5

‘H. In the abstract, the experimental part is suppressed, the calculational aspect is overestimated and presented in the form of the finally proven facts. In the manuscript, the ‘six-fold valley degeneracy’ is not even explicitly mentioned, e.g. in the comments to the figure 3.

The same impression makes also the summary. Here, the multi-valley approach is presented as a basis for the materials design, which was not the issue in the presented manuscript.’

Reply

In the revised manuscript, the overestimation of theoretical calculation is explained. Six-fold valley degeneracy, which is another expression of a valley degeneracy of 6, is added in the caption of Figure 3. Please see the reply to Comment 4 for details.

Multi-valley band behavior was explained in detail for n-type Mg_3Sb_2 . Several methods, including *ab initio* band structure, Fermi surface, and the dependence of Seebeck coefficient and power factor on carrier concentration, have been used to demonstrate the multiple band behavior in n-type Mg_3Sb_2 . The multiple conduction band feature of $\text{Mg}_3\text{Sb}_{1.5}\text{Bi}_{0.5}$ is similar to that of Mg_3Sb_2 . They have the ML band with a six-fold valley degeneracy and the K band with a two-fold valley degeneracy. The only difference, however, is in $\text{Mg}_3\text{Sb}_{1.5}\text{Bi}_{0.5}$ the ML band moves downwards 0.12 eV below the K band, which makes the ML band easier to reach at a low doping level. To make the multiple band behavior clearer, a few modifications are applied. Please see the reply to Comment 4 for details. Thanks.

Reviewer 2

Comment 1

'In the experimental, the source of the elements should be specified.'

Reply

Thank you. We have added the source of the elements in methods.

On page 13-14 in *Methods*, "High-purity elements Sb pieces (99.9999%, **Chempur**), Bi pieces (99.999%, **Chempur**) and Te pieces (99.999%, **Sigma Aldrich**) were weighted, ground into powder (<100 μm) in an agate mortar, Mg powder (99.8%, 44 μm , **Alfa Aesar**) in a ball mill mixer for 15 min."

Comment 2

'The only other suggestion would have been to prepare a few more samples as I typically like to see a progression to high zT rather than just one sample. However, I assume that the authors are in the process of preparing more samples for a more complete paper and a communication is warranted with the combination of the theoretical support.'

Reply

Thanks for the suggestion. We agree and that is why we originally submitted the paper based on a single sample. However, we also acknowledge that the study is more definitive when based on multiple samples and as explained above we have in the revised manuscript added data for more samples with different compositions. The progression to high zT can be seen in Fig. 1a in the revised manuscript. Please see the reply to the Comment 1 of the first reviewer for details.

Reviewer 3

Comment 1

'1. There are a variety of zT values depending on the method of sample preparation. In this sense, the authors need to show their own data on pristine Mg_3Sb_2 and $\text{Mg}_3\text{Sb}_{1.5}\text{Bi}_{0.5}$ samples, which need to be compared with their own theoretical results for p type cases.'

Reply

Thank you for the suggestion. We agree that the zT value depends on the method of sample preparation, and this is why we carefully chose the reported data (*RSC Adv.* 3, 8504-8516 (2013); *Acta Mater.* 93, 187-193

(2015)) using the similar synthesis method (i.e. Mechanical milling followed by SPS press) for comparison. However, to additionally address the concern, we have prepared one Mg_3Sb_2 sample using our synthesis method. The obtained zT value by our synthesis method is very similar to the reported data cited in the manuscript (see Figure A).

We would like to stress that the focus of this work is on n-type Mg_3Sb_2 -based materials rather than p-type. Therefore, we think using the previous reported data is reasonable and does not affect the present work.

Figure A. zT value of p-type Mg_3Sb_2 by our synthesis method in comparison to the previous reported data using the similar synthesis method.

Comment 2

'2. The authors only show the data of $\text{Mg}_3\text{Sb}_{1.5-0.5x}\text{Bi}_{0.5-0.5x}\text{Te}_x$ ($x=0.05$). Even though I do agree that n-type doping is effective using tellurium, it is more persuasive to show the data of $\text{Mg}_3\text{Sb}_{1.5}\text{Bi}_{0.5}$ with various Te dopants, x . It gives more convincing n_H dependence, for example more data points in Figs. 2a, 2b, and 3d.'

Reply

Thank you for the comment. As explained above we have made more samples with different compositions and included the data in the revised manuscript. Please see the reply to the Comment 1 of the first reviewer.

Comment 3

'3. The authors mentioned that n-type doping in Sb sites is a lot easier in $\text{Mg}_3\text{Sb}_{1.5}\text{Bi}_{0.5}$ solid solutions. This opinion needs to be demonstrated ei-

ther experimentally (maybe, based on TEM or XPS?) or theoretically (maybe, based on formation energy or electronegativity?). Moreover, I doubt the nominal composition $\text{Mg}_3\text{Sb}_{1.5-0.5x}\text{Bi}_{0.5-0.5x}\text{Te}_x$ ($x=0.05$). According to their opinion, $\text{Mg}_3\text{Sb}_{1.5}\text{Bi}_{0.5-x}\text{Te}_x$ is more effective?’

Reply

Following the suggestion by the reviewer we have proven our point by calculating the formation energy of Te doping on the anion sites in Mg_3Sb_2 and $\text{Mg}_3\text{Sb}_{1.5}\text{Bi}_{0.5}$. The calculation details are shown in Supplementary Note 6. The result is shown in Supplementary Fig. 2. As shown in the figure, doping tellurium on the anion sites (Sb or Bi sites) in $\text{Mg}_3\text{Sb}_{1.5}\text{Bi}_{0.5}$ has lower formation energy than that in Mg_3Sb_2 . We prepared our samples with the nominal compositions $\text{Mg}_3\text{Sb}_{1.5-0.5x}\text{Bi}_{0.5-0.5x}\text{Te}_x$ ($x=0.04, 0.05, 0.08, 0.20$) since initially we do not know if Te will prefer to substitute Sb atoms or Bi atoms. In addition, quantitative elemental analysis of the high-performance pellet with $x = 0.05$ was carried out by SEM-EDS (see Supplementary Table 1). We find that the actual composition is close to the nominal composition, and the actual composition shows slightly less Bi and more Sb relative to the nominal composition. This can be explained by the assumption that more Te has been doped on the Bi sites than on the Sb sites due to the lower formation energy of Te_{Bi} . The above result indicates that in the future we could try synthesis of n-type $\text{Mg}_3\text{Sb}_{1.5}\text{Bi}_{0.5-x}\text{Te}_x$, which might be more effective. We have made the following revisions to address this comment:

- (a) A few sentences are added in the beginning of second paragraph on page 9: “Achieving n-type Mg_3Sb_2 by doping tellurium on the anion sites has been attempted and found to be difficult, while it is easier in $\text{Mg}_3\text{Sb}_{2-x}\text{Bi}_x$ solid solutions. This is probably due to the lower formation energy of tellurium doping on the anion sites of $\text{Mg}_3\text{Sb}_{2-x}\text{Bi}_x$ solid solutions (see one example in Supplementary Fig. 2).”
- (b) A few sentences are added in the second paragraph on page 14: “Quantitative elemental analysis of the high-performance pellet with $x = 0.05$ was carried out on FEI Nova Nano SEM 600 equipped with an element EDS X-ray detector. The result shown in Supplementary Table 1 was the average value from five randomly selected areas of the pellet.”
- (c) Supplementary Fig. 2 and Supplementary Table 1 are added to the Supplementary Information. Supplementary Table 1 has the caption: “**Supplementary Table 1.** The actual composition estimated by SEM-EDS analysis for the high-performance $\text{Mg}_3\text{Sb}_{1.5-0.5x}\text{Bi}_{0.5-0.5x}\text{Te}_x$ sample with $x = 0.05$. The actual composition by SEM-EDX is close to the nominal composition. The actual composition shows less Bi and more Sb

relative to the nominal composition. This is possibly because that more Te has been doped on the Bi sites than on the Sb sites due to the lower formation energy of Te_{Bi} (see Supplementary Fig. 2).”

(d) The calculation details of defect formation energy are provided and added as Supplementary Note 6 to Supplementary Information.

Comment 4

‘4. I believe that the authors did not consider temperature effect in their DFT calculations. Nevertheless, they compare the theoretical data of n-type Mg_3Sb_2 with the experimental data taken at 300 K (see Figs. 2a and 2b), and the theoretical data of n-type $\text{Mg}_3\text{Sb}_{1.5}\text{Bi}_{0.5}$ with the experimental data taken at 725 K (see Fig. 3d). If one considers different temperature, for example, 300 K in Fig. 3d and 725 K in Figs. 2a and 2b, the experimental data do not agree with the simulated curves.’

Reply

This appears to be a misunderstanding. The temperature effect is included in the electrical transport calculation by semi-classical Boltzmann transport theory under the constant scattering time approximation (CSTA). Under CSTA, it is assumed that the carrier scattering time τ determining the electrical conductivity will not vary strongly with temperature and doping level. This approach has been successfully applied to predict the Seebeck coefficient and the trend of the electrical conductivity or power factor for a variety of thermoelectric materials (*J. Am. Chem. Soc.* 128, 12140-12146 (2006); *Adv. Funct. Mater.* 18, 2880-2888 (2008); *Adv. Mater.* 26, 3848-3853 (2014).). Please see Supplementary Note 3 for the details on electrical transport calculations.

Concerning the reason why the DFT curve looks overestimated. This is because the DFT curve is simulated for n-type Mg_3Sb_2 with nearly converged ML band and K band, while the experimental data is for n-type $\text{Mg}_3\text{Sb}_{1.5}\text{Bi}_{0.5}$, which has the ML band lies 0.12 eV below the K band. Please see the reply to Comment 4 of the first reviewer for details. Thanks.

Comment 5

‘5. According to their n_{H} and μ_{H} data; when simple looking, n_{H} is almost 2 times increased with temperature and μ_{H} is almost 3 times decreased with temperature, the resistivity should be increased about 1.5 times, which seems to be inconsistent with Fig. 4c.’

Reply

The inconsistency is caused by different resistivity measurement methods between our homebuilt setup and the commercial ZEM-3 setup. The mobility was calculated from the Hall coefficient and resistivity measured on the full pellet using four-point *Van der Pauw* method in our home built setup (*Rev. Sci. Instrum.* 2012, 83, 123902.). The resistivity shown in Fig. 4c was measured on the bar sample using the commercial ZEM-3 setup with the DC four-terminal method. For comparison of different transport measurement techniques, please see *Energy Environ. Sci.* 2015, 8, 423-435. The resistivity values, measured by two different methods on samples with different geometries, are normally inconsistent and this will cause some uncertainties. But the uncertainties are within the inter-laboratory value $\pm 6.5\%$ from the International Round-Robin Study (see *J. Electron. Mater.* 2015, 44, 4482-4491.). However, in order to fully address the inconsistency, we have measured all high-temperature thermoelectric properties including Seebeck coefficient, resistivity, Hall data, and thermal diffusivity on the same pellet using our homebuilt setups to reduce uncertainties, and the data are shown in the revised main text. The data measured on the ZEM-3 setup are shown in the Supplementary Information for comparison. The revisions are summarized below:

- (a) Figures 1-4 in the main text are modified using the data measured by our homebuilt setups on the same pellet.
- (b) A few words have been added to the third paragraph on page 14: “The in-plane Hall coefficient (R_H) and resistivity ρ were measured on the pellets using the *Van der Pauw* method in a magnetic field up to 1.25 T (ref. 29).”
- (c) One sentence is added in the beginning of page 15: “The Seebeck coefficients of the pellets were then measured from the slope of the thermopower versus temperature gradient using chromel-niobium thermocouples on an in-house system, which is similar to the one reported by Iwanaga et al³⁰.” In addition, one reference is added: “30. Iwanaga, S., Toberer, E. S., LaLonde, A. & Snyder, G. J. A high temperature apparatus for measurement of the Seebeck coefficient. *Rev. Sci. Instrum.* **82**, 063905 (2011).”
- (d) A few words are added in line 9 of page 15: “For a comparison, one high-performance pellet $Mg_3Sb_{1.5-0.5x}Bi_{0.5-0.5x}Te_x$ ($x = 0.05$) was polished and cut into a $2 \times 2 \times 9$ mm bar for the measurement of electrical transport properties, including electrical resistivity (ρ) and Seebeck coefficient (α), using a ZEM-3 (ULVAC) apparatus under a helium atmosphere from 300 to 725 K (Supplementary Fig. 8 and Supplementary Note 2).”

(e) One sentence is added in line 13 of page 15: “The thermoelectric zT , obtained on the high-performance sample with $x = 0.05$ by the home-built system and ZEM-3 setup, shows good consistency between each other (Supplementary Fig. 9).”

Comment 6

‘6. Also, the Seebeck coefficient at 725 K in Fig. 4b seems to be about 280 microV/K, but the data point in Fig. 3d is about 300 microV/K. They are not consistent each other.’

Reply

We have checked the Seebeck coefficients at 725 K in Fig. 4b and Fig. 3d in the initial version of manuscript and confirmed that they are consistent with each other (282 microV/K). Please note that the scales of Fig. 4b and Fig. 3d are not the same.

As explained above in the revised manuscript, we have used the electrical transport data measured on our homebuilt setups in the main text. The ZEM-3 data have been put in the Supplementary Information for comparison (see the reply to Comment 5).

Comment 7

‘7. The authors need to explain the reason on thermal hysteresis of Seebeck coefficient and electrical resistivity (Figs. S7a and S7b). If it originates from annealing effect and/or slight oxidation, the hysteresis may not be reversible.’

Reply

Although the transport properties show hysteresis, the hysteresis can be repeated very well in two different cycles (see Supplementary Fig. 8). Moreover, the thermoelectric zT obtained on the high-performance sample with $x = 0.05$ by the home-built system and the commercial ZEM-3 setup shows excellent consistency between each other (Supplementary Fig. 9). The good repeatability and reproducibility of the high zT observed in two different setups indicate that the hysteresis in the sample is likely caused by reversible processes, and that it is not due to an annealing effect or oxidation. If the hysteresis is caused by annealing or oxidation, the transport properties curves will not be repeated in two different cycles and the high zT will not be reproduced by two different setups.

We stress that the focus of the present work is proof-of-concept of multi-valley conduction bands in n-type Mg_3Sb_2 -based compounds. We have

demonstrated that the hysteresis does not affect the high performance in the n-type sample, and exploring the exact the reason of thermal hysteresis is beyond the scope of the present work. We are planning to perform *in-situ* X-ray diffraction on n-type samples for a comprehensive study of structure-property relation, which might explain the hysteresis in the future.

- (a) The comparison of zT values measured on ZEM-3 and homebuilt setup is plotted in Supplementary Fig. 9. Supplementary Fig. 9 is added in the Supplementary Information.
- (b) A few sentences are added to the Supplementary Note 2: “The transport properties of the two cycles are consistent upon the repeated heating and cooling measurements. **In addition, the thermoelectric zT , obtained on the high-performance sample with $x = 0.05$ by the home-built system and ZEM-3 setup, shows excellent consistency between each other (Supplementary Fig. 9). The good repeatability and reproducibility observed in different setups indicate that the hysteresis in the sample is more likely caused by reversible processes.**”

Summary of the changes (marked in red in the revised manuscript)

1. Two words are modified in line 5 and a few words are added in line 8 of the abstract.
2. A few words are modified in line 3-10 of second paragraph on page 3.
3. A few words are modified in the end of page 7.
4. A few words are added in line 5 of second paragraph on page 8.
5. One word is modified in line 7, two words are added in line 8, and a few words are added in line 10 of second paragraph on page 8.
6. Two sentences are added in the beginning of page 9.
7. Two sentences are added in the beginning of the second paragraph on page 9.
8. Two sentences are updated and added in the end of the second paragraph on page 9.
9. A few sentences are modified in the end of page 9 and in the beginning of page 10.
10. The first sentence of the second paragraph on page 10 is modified.
11. A few words are added in line 8 of the second paragraph on page 10.

12. A few words are modified in the end of the second paragraph on page 10.
13. A few sentences are added in the end of page 10.
14. A few words are modified in line 2 of page 11.
15. A few words are modified in line 4-6 of page 11.
16. Two sentences are added in line 7-11 of page 11.
17. A few words are modified in line 2 of the second paragraph on page 11.
18. A few sentences are added in the end of page 11.
19. A few sentences are modified in the beginning of page 12.
20. A few words are modified in line 6-8 of page 12.
21. A few words are added in the end of the first paragraph on page 12.
22. Three sentences are modified in the beginning of the second paragraph on page 12.
23. A few sentences are modified in the end of page 12 and in the beginning of page 13.
24. A few words are added in the final paragraph of page 13.
25. A few words are added in the first paragraph of page 14.
26. Two sentences are added in the end of second paragraph of page 14.
27. A few words are added in the beginning of the third paragraph of page 14.
28. One sentence is added in line 2-5 of page 15.
29. A few words are added in line 6 and line 9 of page 15.
30. One sentence and a few words are added in the end of the first paragraph on page 15.
31. A few words are added in line 4 and line 8 of page 16.
32. The final sentence of the second paragraph on page 16 is modified.
33. One reference is added on page 19.
34. Figures 1, 2a,b, 3b,c,d, and 4 are modified. The order of Figure 3c and 3d is changed.
35. A few words are modified in the caption of Figure 1.
36. One sentence is modified and one sentence is added in the caption of Figure 2a,b.

37. Two sentences are added in the caption of Figure 3a. One sentence is modified in the caption of Figure 3b. The captions of Figure 3c and Figure 3d are swapped.
38. A few words are modified in the caption of Figure 4.
39. The order of the figures in Supplementary Information is changed.
40. Three new figures including Supplementary Figs 2, 4, and 9 are added in Supplementary Information.
41. A new table and a new note (Supplementary Table 1 and Supplementary Note 6) are added in Supplementary Information.
42. The order of Supplementary Notes is changed.
43. Supplementary Figs 3, 5, 6, and 7 and their captions are modified and updated.
44. A few sentences and words are modified and updated in Supplementary Notes 1, 2, 3, 4, and 5.
45. Five references are added in the References of Supplementary Information.

Once again we would like to thank the reviewers for the time they have devoted to our manuscript. Their very insightful comments have significantly improved our manuscript and we look much forward to your decision.

REVIEWERS' COMMENTS:

Reviewer #1 (Remarks to the Author):

In principle, the authors addressed the most remarks of the reviewer. Nevertheless, the manuscript is still suffering on the very technical presentation which is not very perceptible for the wide audience. Especially the theoretical part where the authors are using mostly traditional techniques is still overestimated.

In contrary, the aspects which would be convincing for wide audience are only fragmentarily presented. E.g. the title claims high-performance low-cost materials, the performance of Mg₂Si-Mg₂Sn material is very much comparable with the title one (this is also not commented), but the costs are definitely lower.

Reviewer #2 (Remarks to the Author):

An excellent paper, combining both theory and experiment to obtain a high zT material. All of my previous questions/comments are well addressed and the authors have improved the manuscript.

Reviewer #3 (Remarks to the Author):

Now, the manuscript is reasonably revised according to the reviewers' comments. So, I would suggest it to be published.

Thank you for the three referees' reports. We are grateful for the constructive and helpful comments from the referees, and below we address the reports point by point.

Reviewer 1

Comment 1

'In principle, the authors addressed the most remarks of the reviewer. Nevertheless, the manuscript is still suffering on the very technical presentation which is not very perceptible for the wide audience. Especially the theoretical part where the authors are using mostly traditional techniques is still overestimated.

In contrary, the aspects which would be convincing for wide audience are only fragmentarily presented. E.g. the title claims high-performance low-cost materials, the performance of Mg₂Si-Mg₂Sn material is very much comparable with the title one (this is also not commented), but the costs are definitely lower.'

Reply

The overestimation of theoretical calculation was explained in detail in the previous response letter. In addition, we also explained the reason of overestimation in the revised manuscript.

Mg₃Sb₂-based Zintl compound is a "low-cost" material compared with the commercial Bi₂Te₃ and most Zintl compounds. We agree that the cost of Mg₂Si-Mg₂Sn with comparable performance is lower. This material has been widely studied for a long time. However, it shows serious obstacles for application, such as phase decomposition and degradation of proper-

ties with oxidation. N-type Mg_3Sb_2 -based Zintl compound, as a relatively new material, could be competing with Mg_2Si - Mg_2Sn and become a very promising candidate for applications.

Once again we would like to thank the reviewers for the time they have devoted to our manuscript. Their very insightful comments have significantly improved our manuscript and we look much forward to your decision.